# Molecular characterization of carbapenem-resistant *Acinetobacter baumannii* clinical isolates from Egyptian patients

**Reem M. Hassan[1], Sherifa T. Salem[1], Saly Ismail Mostafa Hassan[2], Asmaa Sayed Hegab[3]\*, Yasmine S. Elkholy[3]**

**1** Faculty of Medicine, Department of Clinical and Chemical Pathology, Cairo University, Cairo, Egypt, **2** Faculty of Medicine, Department of Clinical Pathology, Beni-Suef University, Beni-Suef, Egypt, **3** Faculty of Medicine, Department of Medical Microbiology and Immunology, Cairo University, Cairo, Egypt

\* asmahgab@kasralainy.edu.eg

**Data Availability Statement:** All relevant data are within the manuscript and its Supporting information files.

## Abstract

*Acinetobacter baumannii* (*A. baumannii*) represents a global threat owing to its ability to resist most of the currently available antimicrobial agents. Moreover, emergence of carbapenem resistant *A. baumannii* (CR-AB) isolates limits the available treatment options. Enzymatic degradation by variety of ß-lactamases, have been identified as the most common mechanism of carbapenem resistance in *A. baumannii*. The alarming increase in the prevalence of CR-AB necessitates continuous screening and molecular characterization to appreciate the problem. The present study was performed to assess the prevalence and characterize carbapenemases among 206 CR-AB isolated from various clinical specimens collected from different intensive care units at Kasr Al-Aini Hospital. All isolates were confirmed to be *A. baumannii* by detection of the *bla*OXA-51-like gene. Molecular screening of 13 common Ambler class bla carbapenemases genes in addition to insertion sequence (IS-1) *upstream* OXA-23 were performed by using four sets of multiplex PCR, followed by identification using gene sequencing technology. Among the investigated genes, the prevalence of *bla*OXA-23, and *bla*OXA-58 were 77.7%, and 1.9%, respectively. The IS*Aba1* was detected in 10% of the *blaOXA*-23 positive isolates. The prevalence of metallo-β-lactamases (MBLs) studied; *bla*NDM-1, *bla*SPM, *bla*VIM, *bla*SIM-1 were 11.7%, 6.3%, 0.5%, and 0.5% respectively. One of class A; *bla* KPC was detected in 10.7% of the investigated isolates. *bla*OXA-24/40, *bla*IMP, *bla*GES, *bla*VEB and *bla*GIM were not detected in any of the studied isolates. Moreover, 18.4% of the isolates have shown to harbor two or more of the screened *bla* genes. We concluded that the most prevalent type of ß-lactamases genes among CR-AB isolates collected from Egyptian patients were *bla*OXA-23 followed by *bla*NDM-1 and *bla*KPC.

## Introduction

*A. baumannii* has been identified as an opportunistic nosocomial pathogen, which is tolerant to wide ranges of temperature, pH, and humidity and is able to survive on dry surfaces for

**Funding:** The author received no specific funding for this work.

**Competing interests:** The authors have declared that no competing interests exist.

months, so it is recognized as a real challenge to infection control measures [1]. It causes a wide variety of infections including pneumonia, urinary tract infection, skin and soft tissue infections, central line associated bloodstream infections and others [2].

*A. baumannii* represents a global threat owing to its ability to resist most of the currently available antimicrobial agents including ß- lactams, fluoroquinolones and aminoglycosides [3]. Un-regulated use of antibiotics in healthcare settings results in emergence of multidrug-resistance added to the intrinsic resistance of *A. baumannii* especially in intensive care units [4]. Moreover, emergence of carbapenem resistant *A. baumannii* isolates limit the available treatment options for such infections [5].

The most common mechanism responsible for carbapenem resistance among *A. bauminii* is enzymatic degradation by variety of ß-lactamases, the four β-lactamase classes (A, B, C and D) have been detected in *A. baumannii* [6–8].

Two intrinsic types of ß-lactamases can be identified in almost all *A. baumannii* isolates: AmpC-type cephalosporinases and OXA-51/69 variants. Both of which are chromosomally located, and have little impact on carbapenems, if any [9].

However, higher carbapenem hydrolysis rates have been reported to result from the acquisition of insertion sequences (ISs) that affect the expression of *bla*OXA genes encoding oxacillinases [10]. These elements are the most abundant transposable elements capable of causing mutations and rearrangements in the genome, contributing to the spread of resistance and virulence determinants among bacterial species [11]. Insertion sequence *A. baumannii*-1 (IS*Aba1)* upstream *bla*OXA-51 like genes, belong to the IS4 family and has been associated with increased gene expression and hence carbapenem resistance [8,12].

Till date, carbapenem-hydrolyzing class D β-lactamases, the so called (CHDLs) also named oxacillinases (OXAs) for their effect on oxacillin [13] seem to be the most common mechanism of carbapenem resistance in *A. baumannii* [8]. Class D β-lactamases, includes six subgroups.

The intrinsic OXA-51 and the acquired OXA-23-like, OXA-58-like, OXA-24/40-like, OXA-235-like and OXA-143-like β-lactamases [13].

Additionally, class B β-lactamases, also known as MBLs play a less important yet more potent role in carbapenem resistance among *A. baumannii* isolates [14].

Four MBLs are known in *A. baumannii*: New Delhi metallo-β-lactamase (NDM), Imipenemase (IMP), Seoul Imipenemase (SIM) and Verona integron-encoded metallo-β-lactamase (VIM) [14,15]. Two variants of NDM (NDM-1 and NDM-2) have been reported in *A. baumannii* clinical isolates in Egypt [16,17].

Moreover, β-lactamases belonging to class A including *bla*KPC and *bla*GES were also incriminated of carbapenem resistance among *A. baumannii* [18].

Other ESBLs were identified among *A.baumannii* isolates and confer resistance to broad spectrum cephalosporins e.g. *bla*PER, *bla*VEB and others [19].

Despite of the global increase at alarming rates of CR-AB, few studies were devoted to this organism in Egypt [20–22]. More data are crucial to appreciate the problem objectively and to control further evolution of CR-AB.

The aim of the present study was to characterize and to assess the prevalence of carbapenemases among 206 CR-AB clinical isolates from Egyptian patients.

## Materials and methods

Being a teaching hospital, before admission every patient gives an informed consent. Samples were collected for diagnostic purposes and were furtherly investigated for better analysis of antimicrobial resistance to allow control. This cross- sectional study involved a total of 206 clinical, non-duplicate, isolates of CR-AB which were collected over one year duration from

December 2018 to December 2019. The research study was approved by the institutional Review Board of Kasr Al-Aini Hospital and the Research Ethics Committee at Faculty of Medicine, Cairo University.

## I. Bacterial isolates

The isolates were collected at the central microbiology laboratory at Kasr Al-Aini Hospital. The isolates were obtained from different clinical specimens, including wound swabs, respiratory secretions, blood cultures, urine samples and others including body fluid and drains collected from inpatients admitted at intensive care units and other different departments at Kasr Al-Aini Hospital.

The isolates were identified phenotypically by colonial morphology, Gram- staining and conventional biochemical testing. All isolates grown as lactose non fermenter colonies on MacConkey agar, appearing Gram negative coccobaciili were preliminary identified as *Acinetobater* spp. [23].

## II. Carbapenem susceptibility testing

Carbapenem susceptibility testing for the initially identified *Acinetobacter* clinical isolates was done by the standard disc diffusion technique on Müller-Hinton agar using imipenem and meropenem discs (Oxoid, Basingstoke, United Kingdom), and interpreted following the Clinical and Laboratory Standards Institute (CLSI) [24].

## III. Molecular identification

All CR-AB isolates were selected and submitted to DNA extraction by the heat shock method, followed by genotypic identification of *A. baumannii* by detection of the *bla*OXA-51-like gene [25]. The 206 isolates were confirmed to be *A. baumannii* were further investigated.

## IV. Molecular detection of different carbapenemases encoding genes and insertion sequence (IS) *bla*OXA-23

Four sets of multiplex PCR were done including common Ambler class *bla* genes that cause carbapenem resistance; multiplex 1 included *bla*OXA-23, *bla*OXA-24 and *bla*OXA-58 [26,27], multiplex 2 included *bla*VIM, *bla*KPC and *bla*IMP, multiplex 3 included *bla*GES, *bla*PER and *bla*VEB [28], while multiplex 4 contained *bla*GIM, *bla*SIM-1, *bla*SPM and *bla*NDM-1 [28,29]. Presence of IS*Aba1* upstream of *bla*OXA-23 was investigated using IS*Aba1*F/OXA-23-likeR [12]. PCR was performed using PCR-EZ D-PCR Master Mix (Bio Basic Inc., Canada) in a Bio-Rad Thermal Cycler PTC-200. Briefly, an initial denaturation step of 95˚C for 15 min, followed by 30 cycles of denaturation at 94˚C for 30 sec, an annealing temperature dependent on the melting temperature of the primer pair (multiplex I and IS*Aba1*F/OXA-23-likeR: 52˚C; multiplex II and III: 57˚C and multiplex IV: 60˚C) and extension at 72˚C for 90 sec, followed by the final extension step at 70˚C for 10 min. A negative control (sterile nuclease free water) was included for all PCR assays. Amplified PCR products were purified using PureLink® PCR Purification Kit (Invitrogen, Carlsbad, CA, USA) according to manufacturer's instructions. A BigDye Terminator v3.1 cycle sequencing kit (Applied Biosystems, Foster City, CA, USA) was used to sequence the PCR amplified products from the positive cases according to manufacturer's instructions. The sequenced products were run on a 3500 Genetic Analyzer (Applied Biosystems). Sequences were compared with those available in the GenBank database using the basic local alignment search tool (BLAST, www.ncbi.nlm.nih.gov).

**Table 1. Distribution of *A. baumannii* isolates among different clinical specimens.**

| Clinical Specimen | *A. baumannii* isolates N (%) |
|---|---|
| Wound discharge | 77 (37.4%) |
| Respiratory secretions | 56 (27.2%) |
| Blood | 37 (18.0%) |
| Urine | 27 (13.1%) |
| Others (body fluids, drains) | 9 (4.4%) |

## Results

A total of 206 clinical isolates of CR-AB were collected from different clinical specimens mainly from different ICUs (n = 154, 74.8%) at Kasr Al-Aini Hospital. Most of the CR-AB detected were isolated from wound swabs (n = 77), respiratory secretions (n = 56), blood cultures (n = 37), and urine samples (n = 27) (Table 1).

All isolates were confirmed to be *A. bauminii* by detecting *bla*OXA-51-like gene among all (100%) the studied isolates (Table 2).

Among the investigated genes, class D *bla*OXA-23 was the most commonly detected gene (77.7%) and the *ISAba1* was detected only in 10% of the *bla*OXA-23 positive isolates.

One of the MBLs; *bla*NDM was detected in 11.7%, and one of class A; *bla*KPC was detected in 10.7% of the investigated isolates. Other carbapenemases were detected at lower frequencies. However, five of the investigated genes (*bla*OXA-24, *bla*IMP, *bla*GES, *bla*VEB and *bla*GIM) were not detected in any of the studied isolates (Table 2).

Although most of the studied CR-AB isolates were detected in specimens from different ICUs, there was no significant difference in the prevalence of the investigated genes in clinical specimens collected from the ICUs and specimens collected from other departments (Table 3).

Moreover, 18.4% of the isolates (n = 38) have shown to harbor two or more of the tested *bla* genes (Table 4).

**Table 2. Distribution of different bla gene classes and genes among the CR-AB isolates.**

| *bla* class | *bla* gene | *A. baumannii* isolates N (%) |
|---|---|---|
| Class A | *bla*KPC | 22 (10.7%) |
| | *bla*GES | 0 (0.0%) |
| Class-B | *bla*VIM | 1 (0.5%) |
| | *bla*IMP | 0 (0.0%) |
| | *bla*GIM | 0 (0.0%) |
| | *bla*SPM | 13 (6.3%) |
| | *bla*SIM-1 | 1 (0.5%) |
| | *bla*NDM-1 | 24 (11.7%) |
| Class D | *bla*OXA-51 | 206 (100%) |
| | *bla*OXA-23 | 160 (77.7%) |
| | *bla*OXA-24/40 | 0 (0.0%) |
| | *bla*OXA-58 | 4 (1.9%) |
| Other beta-lactamases | *bla*PER | 2 (1.0%) |
| | *bla*VEB | 0 (0.0%) |
| ISAb1 upstream of *bla*OXA-23 | | 16 (7.8%) |

**Table 3. Difference in distribution of bla genes among CR-AB isolates from ICU and non-ICU clinical specimens.**

| *bla* gene | ICU- Clinical specimens N (%) | Non-ICU Clinical Specimens N (%) | P-value |
|---|---|---|---|
| **OXA-23** | 125 (81.2%) | 35 (67.3%) | 0.038 |
| **OXA- 58** | 3 (1.9%) | 1 (1.9%) | 1.000 |
| **VIM** | 1 (0.6%) | 0 (0.0%) | 1.000 |
| **KPC** | 13 (8.4%) | 9 (17.3%) | 0.073 |
| **PER** | 1 (0.6%) | 1 (1.9%) | 0.442 |
| **SPM** | 12 (7.8%) | 1 (1.9%) | 0.191 |
| SIM-1 | 1 (0.6%) | 0 (0.0%) | 1.000 |
| **NDM-1** | 16 (10.4%) | 8 (15.4%) | 0.332 |
| **ISABA1** | 10 (6.5%) | 6 (11.5%) | 0.240 |

## Discussion

Outlining the genetic background of carbapenem resistance among 206 *A. baumannii* isolates collected from Kasr Al Aini Hospital, one of the largest tertiary care hospitals in Egypt, was the main scope of the present study. As proved by other studies, *bla* OXA-51, is an oxacillinase present in all *A. baumannii* isolates. Luckily, it is too weak to confer by itself carbapenem resistance. However, it is frequently used to confirm the identity of *A.baumannii* [27,30]. All the investigated isolates of our study were confirmed to harbor this gene.

In concordance with other studies, the most common carbapenemase gene was *bla*OXA-23 that belongs to Class D β-lactamases. In the present study, it was detected in 77.7% of the investigated isolates. In Egypt, a study in Zagazig University Hospitals reported that 90% of 50 carbapenem resistant *A. baumannii* isolates were found to harbor *bla*OXA-23 [31]. Obviously, the higher carriage rate in the later study may be attributed to the fact that all isolates were collected from surgical ICUs, whereas in our study the investigated isolates were isolated from both ICU and non-ICU settings. Other studies in Egypt and Saudi Arabia showed higher prevalence rate (100%) of *bla*OXA-23 among carbapenem resistant *A. baumannii* isolates [32,33]. IS*Aba1* was detected upstream *bla*OXA-23 gene only in 10% of the *bla*OXA-23 positive isolates. These findings confirmed that the *bla*OXA-23 gene could be located on the chromosome or on a plasmid [34]. We can conclude that the current worldwide dissemination of the *bla*OXA-23 gene is associated with different genetic structures and plasmids. The dynamic

**Table 4. CR-AB isolates harbor more than one of the bla genes.**

| *bla* genes | CR-AB isolates harbor more than one *bla* genes (N = 38) |
|---|---|
| OXA-23, KPC | 10 |
| OXA-23, NDM | 6 |
| OXA-23, SPM | 6 |
| OXA23, KPC, NDM | 6 |
| OXA-58, NDM | 3 |
| OXA-23, PER | 2 |
| OXA-23, NDM, SPM | 2 |
| OXA-23, SIM | 1 |
| OXA-23, OXA-58, KPC | 1 |
| OXA-23, NDM, VIM | 1 |

spread of *bla*OXA-23 will make it difficult to control because this spread is not associated with a single entity [35].

Within the same class, class D β-lactamases, *bla*OXA-58 was detected only among 1.9% of our studied isolates. Closely related to our results, other studies in Egypt, in USA, and in Palestine could detect *bla*OXA-58 among 1.4%, 2%, and 3% respectively among carbapenem resistant *A. baumannii* investigated isolates [22,36,37]. However, another Egyptian study could not find the former gene among any of their studied carbapenem resistant *A. baumannii* isolates [38]. Although, a higher *bla*OXA-58 prevalence was reported among carbapenem resistant *A. baumannii* isolates in Egypt (9.1%) [39] and Algeria (14.7%) [40]. Therefore, more studies are needed from different region in Egypt and moreover from different countries before we can conclude that the prevalence of *bla*OXA-58 is low among carbapenem resistant *A. baumannii*.

Till date the most common carbapenemsases are carbapenem-hydrolyzing class D β-lactamases (CHDLs) and, to a lesser extent, MBLs. Although MBLs has been reported in sporadic parts of the world [41], most of MBLs were first described in *A. baumannii* in Egypt [16,17,42] then emerged in the middle east [43]. In the present study MBLs were detected in 19% of the investigated isolates; with *bla*NDM accounts for 11.7%, *bla*SPM 6.3%, *bla*VIM 0.5%, and *bla*SIM 0.5%. Nevertheless, in the present study, none of the *A. baumannii* isolates were found to harbor *bla*IMP, nor *bla*GIM.

In Egypt, many authors have reported different prevalence of MBLs in *A. baumannii*. *Elkasaby and El Sayed Zaki, (2017)* reported that among 280 *A. baumannii* isolates collected from Egyptian patients admitted to Mansoura University Hospital ICU, 95% harbored MBLs, of which, *bla*IMP accounted for 95.7%. However, the authors did not investigate *bla*NDM among the studied isolates [21]. Another Egyptian study has detected *bla*NDM among 66.7% of 50 *A.baumannii* isolates. Yet, the former study didn't find neither *bla*IPM nor *bla*VIM among the tested isolates [31]. Another study in Egypt has shown that *bla*NDM accounted for carbapenem resistance in 27.58%, and *bla*VIM in 10.3% of studied isolates [20]. Two MBLs: *bla*VIM and *bla*NDM were detected in 100% and 12.1% respectively in seventy-four CR-AB investigated isolates, collected from different clinical specimens at Alexandria University Hospital [32]. In Palestine Handal and colleagues reported that *bla*NDM was detected among 5.8% of 69 carbapenem resistant *A. baumannii* isolates [37].

In the present study, two of the Class A carbapenemases, namely *bla*KPC and *bla*GES were tested. While none of isolates were shown to harbor *bla*GES, *bla*KPC was detected in 10.7% of the investigated isolates. Opposing results were detected in other Egyptian studies; *bla* KPC was not detected among 40 carbapenem resistant *A. baumannii* isolates collected from 2 hospitals in Egypt [22,32], while *bla*GES was detected among 50% of the investigated isolates in the later same study [31].

## Conclusion

We can conclude that the worldwide spread of carbapenem-resistant *A. baumannii* has become a real global health threat. Among the investigated isolates in the present study CR-AB were confirmed by detection of *bla*OXA-51. Class D carbapenemase blaOXA-23 was the most prevalent followed by *bla*NDM-1 belonging to class B MBLs and class A *bla*KPC. In this study, IS*Aba1* was detected upstream 10% of *bla*OXA-23 positive isolates only which indicates that the spread of resistance among *Acinetobacter* isolates could be either chromosomal or plamid-mediated. Further investigations should be continuing to appreciate the reality of the problem of multi-drug resistant *A. baumannii*.

## Supporting information

**S1 File.**
(XLS)

**S2 File.**
(XLS)

**S3 File.**
(XLS)

## Author Contributions

**Conceptualization:** Reem M. Hassan, Sherifa T. Salem, Saly Ismail Mostafa Hassan, Yasmine S. Elkholy.

**Data curation:** Reem M. Hassan.

**Formal analysis:** Sherifa T. Salem, Saly Ismail Mostafa Hassan.

**Investigation:** Sherifa T. Salem, Saly Ismail Mostafa Hassan.

**Methodology:** Reem M. Hassan, Sherifa T. Salem.

**Resources:** Asmaa Sayed Hegab.

**Supervision:** Reem M. Hassan, Saly Ismail Mostafa Hassan, Asmaa Sayed Hegab, Yasmine S. Elkholy.

**Validation:** Asmaa Sayed Hegab.

**Writing – original draft:** Yasmine S. Elkholy.

**Writing – review & editing:** Asmaa Sayed Hegab, Yasmine S. Elkholy.

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
